# Language Ideologies, Practices, and Kindergarteners' Narrative Macrostructure Development: Crucial Factors for Sustainable Development of Early Language Education

**Jing Yin** [1], **Yan Ding** [1] and **Lin Fan** [2,*]

1 School of Language and Communication Studies, Beijing Jiaotong University, Beijing 100044, China; Yinj@bjtu.edu.cn (J.Y.); Yanding@bjtu.edu.cn (Y.D.)
2 National Research Center for Foreign Language Education/Artificial Intelligence and Human Languages Lab, Beijing Foreign Studies University, Beijing 100081, China
* Correspondence: fanlin@bfsu.edu.cn

**Abstract:** This paper explored crucial factors to achieve sustainable development of early language education by examining the relationship between two dimensions of family language policy—language ideologies and language practices—as well as the relationship between family language policy and the development of children's narrative macrostructure. Data were collected via a language performance test and a questionnaire survey of 131 kindergartners from 10 kindergartens in a Chinese city. Structural equation modeling corroborated the relationship between family language ideologies and family language practices proposed by family language policy theorists. Results showed that family language policy significantly predicted kindergarteners' development of narrative macrostructure. In addition, age was shown to be a significant predictor of narrative macrostructure development, whereas gender was not. Implications for early intervention of children's narrative macrostructure development were discussed.

**Keywords:** family language policy; narrative macrostructure development; sustainable development; early language education

## 1. Introduction

Education is the key to realizing sustainable development goals. It is a two-way process of teaching and learning, in which language plays an irreplaceable role. Research shows that early language development is an important indicator of future academic success [1,2]. Therefore, ways to sustainable development of early language education should be explored and emphasized. Since "family is the cradle of human growth and parents are the first teachers", family language policy (FLP) should be investigated.

FLP, also referred to as "family language planning" and "family language management and planning", is a language planning activity that takes place within the family domain [3] and revolves around language use among family members [4]. Previous studies expounded on the importance of investigating FLP within the theoretical framework of language policy and child language acquisition [3,5]. These studies contributed greatly to our understanding of the role of FLP in children's language development [6,7]. However, scholars in this field mainly focused on language policy within bilingual or multilingual contexts [5,8]. As language policy and its components exist in each identifiable domain [9], the monolingual context of first language should also be taken into consideration.

To examine FLP in the monolingual context and enrich current FLP studies, we empirically tested the relationships between the components of FLP proposed by FLP theorists and investigated the influence of parents' socio-economic status on FLP in monolingual Chinese kindergartener families in our previous study [10]. The present study followed up the previous one and extended the scope of research from what can influence FLP

to what can be influenced by FLP. Specifically, we focused on the relationship between FLP and the development of children's narrative macrostructure in monolingual Chinese kindergartener families so as to obtain certain implications for sustainable education in early language learning.

We chose to examine narrative macrostructure because it was a good indicator of children's language ability and is therefore worthy of investigation. Narrative, also known as storytelling, refers to the ability to express things or events in an organized manner [11]. It is the core component of child language acquisition and key to the transition from spoken language to reading and writing [12]. Narrative macrostructure, which refers to the structural organization of the narrative ability, lies at the heart of narrative ability, and the development of narrative macrostructure can best reflect the maturity of children's narrative ability [13]. Thus, narrative macrostructure can be said to reflect children's overall language development.

In other words, in order to investigate ways to achieve sustainable education of early language learning, the present study aimed at expanding FLP as a field of inquiry to the study of first language acquisition in monolingual context in home domains by exploring how FLP influenced the development of children's narrative macrostructure. To better explain the predictability of child language acquisition, we also examined the correlations of age and gender to the macrostructure development. The present study expanded the research on FLP and child language acquisition in that it explored the relationship between the two in a monolingual context which, as stated, was not a priority in previous research. In addition, our study discussed what kind of ideologies and practices can promote the development of narrative ability, which can provide useful suggestions for children's early language education.

Specific research questions are as follows: (1) Can family language ideology predict language practice? If so, how? (2) Can family language practice predict children's narrative macrostructure development? If so, how? (3) Can age predicate children's narrative macrostructure development? If so, how? (4) Can gender predict children's narrative macrostructure development? If so, how?

## 2. Literature Review

### 2.1. Family Language Policy and Its Three Dimensions

FLP research can reveal parents' language beliefs and their social attitudes and ideas about language and parenting styles and provides a theoretical basis for studying parent-child interaction and children's language development [14]. It includes three components: language beliefs (or language ideologies), language practices, and language management [15,16]. Family language beliefs refer to the language ideologies behind each language policy, that is, "the ideas with which participants and observers frame their understanding of linguistic varieties and map those understandings onto people, events and activities that are significant to them" [17]. In other words, language ideology is the subconscious beliefs and assumptions about the social utility of a specific language in a specific society, reflecting values and patterns rooted in social language and culture [18] and closely related to the implementation of language policies [19,20]. Language practice emphasizes the actual use of language in different contexts for different reasons. Language management refers to efforts made to interfere with or influence language practices, for example, providing language learning resources for children, taking children on trips, and involving children in language classes to promote their language development [15,21].

FLP theorists usually consider language ideologies as the underlying driving force of language management and practices [3,4] and thus a central component in family language policies [22]. They also suggest that language ideologies are shaped by sociolinguistic, socio-cultural, socio-economic, and socio-political contexts. Socio-cultural context refers to the symbolic values associated with language/languages, socio-economic context refers to instrumental (economic) values ascribed to a language, and social-political context refers to the national education or national language policy [4,15]. It is important to point out

that these factors are said to intertwine and jointly influence individual's personal belief system, for example, whether they are in compliance with the national language policy or the national language education policy in terms of what they should do to facilitate children's language development. These theoretical frameworks were mainly constructed and tested via qualitative inquiries. Therefore, more attention should be given to providing quantitative evidence on testifying whether the forces of family language ideologies may exert influence on family language practices.

Moreover, FLP researchers maintain that FLP is best viewed within the framework of language policy and child language acquisition [3,9]. Most existing studies in this line of research focused on the micro-analysis of parent-child verbal interactions and their acquisition differences at home or laboratory settings, aiming at revealing the language learning mechanisms and the conditions necessary for acquisition to occur [23,24]. Some studies also looked at bilingual and multilingual families and communities, analyzing and explaining children's language preferences and proficiency by focusing on their family language planning forms, family types, environments, and functions of different languages and language learning situations and providing implications for language education and language maintenance [25,26]. However, less attention was given to the various forces of family language ideologies that might exert influence on family language practices within the monolingual context for first language acquisition.

### 2.2. Child Language Acquisition and Narrative Ability Development

Child language acquisition/development refers to a process of learning one or more languages in early childhood through a mechanism under certain conditions [27]. Narrative, also called storytelling, refers to the ability of expressing things and events in an organized way [11]. It is both an important part and an important indicator of children's language ability. Narrative discourse is a true reflection of children's language ability, which is closely related to their later literacy development [28]. By assessing children's narrative ability, people may gain an understanding of children's language development and detect barriers of their language development [29]. Children's narrative ability grows with their age and cognitive ability. Early studies demonstrated that children's narrative comprehension began to develop between 3 to 5 years of age, and their narrative production skills quickly developed between 5 to 7 years old [30]. The ability continued to be refined until 9 years old, when it appeared to resemble that of adults [31]. Narrative ability can also influence children's classroom participation and development of social skills, such as the ability to make friends, etc. [32].

Narrative ability can be viewed from three dimensions, i.e., macrostructure, microstructure, and shining benefits, with macrostructure serving as the core of the three [30]. Macrostructure is also termed global structure [33] and pertains to the coherence level of narratives [34,35]. It refers to the structural level organization of narratives. In the present study, we focused only on macrostructure, which is specifically concerned with the schematic organization of the story and how events in the story relate in meaningful ways.

Previous studies on children's narrative ability mainly focused on the impacting factors related to its development, such as cultural background [36,37], parental roles [38], ways of story-reading [37,39,40], age and gender [41], etc. To date, not much was done on the relationship between children's narrative ability development and their families' FLP, especially ideological factors such as parents' goals, attitudes, or intentions [42]. However, born and growing up into a family, children's narrative ability is inevitably influenced by their families' FLP, because FLP results in differences in children's early language learning experiences, which influence their language development and narrative ability.

### 2.3. Narrative Macrostructure and FLP

Children's narrative skills can be acquired through interactions with others [43] and therefore are influenced by children's home literacy environment [44], shaped by FLP. According to Nelson [43], family language practices such as storytelling, knowledge and

experiences sharing may influence the development of children's narrative skills. Many studies revealed the influence of the frequency of literacy activities and the amount of literacy materials on children's macrostructure development [45,46], and others reported the impacting power of the quality of parents' dialogic reading techniques and children's engagement in reading [47], parents' engagement in shared reading activities, as well as parents' frequent reading activities [46].

Notably, Phillips and Lonigan [48] reported frequency of shared reading and literacy activities are related to socio-economic status. Our previous study also showed that socio-economic status predicts Chinese kindergartener families' FLP [10]. In other words, previous research suggests that socio-economic related factors predict FLP, which shapes and constitutes home literacy environment, which in turn influences the development of children's narrative macrostructure.

### 2.4. FLP Research in China

In recent years, FLP research in China showed an overall upward trend. In 2017, Chinese Journal of Language Policy and Planning launched the special issue of family language policy research, and in the same year, "Multilingualism and Families" Academic Conference and Emerging Scholars' Workshop was held in Wuhan, which officially marked the beginning of FLP study in China. International FLP research development [49], FLP theory and study methods [50], and relationships between FLP and children's language acquisition [51,52] were summarized and introduced to China since then. However, many of these studies remained at a level of speculation or introduction, except a few empirical studies focusing on FLP in different groups of people, for example, Chinese middle-class families [53], Chinese urban families [54], rural migrant workers' families [55,56], ethnic minority families [57,58], etc. Therefore, quality empirical FLP studies in China are still insufficient [59]. Most of the studies remain at the level of language policy; little was done to investigate language development under the framework of FLP and the relationship between FLP and children's narrative macrostructure development.

To sum up our literature review, it revealed that, despite the progress that was made in FLP research and research on narrative ability development, there is a paucity of empirical study of children's narrative ability development within the framework of FLP in a monolingual context. Considering that FLP and its components exist in each identifiable domain, including monolingual ones [9], such investigation is necessary and meaningful for both FLP research and research on children's narrative ability development. As a complement to the previous work in this field, the present study was conducted to examine the relationship between FLP and children's narrative ability development in the Chinese monolingual context.

## 3. Research Hypothesis

The present study aimed to explore approaches to achieve sustainable development of early language education by examining the relationship between FLP and the development of monolingual children's narrative ability. Based on the FLP framework and the literature reviewed in the previous section, we hypothesized that:

**Hypothesis 1 (H1).** *A family's language ideology significantly predicts its language practice.*

**Hypothesis 2 (H2).** *A family's language practice significantly predicts the child's narrative macrostructure performance.*

Considering that age [60] and gender [61,62] were shown to be highly predictive of children's language development, we also took age and gender into consideration and hypothesized that:

**Hypothesis 3 (H3).** *Age significantly predicts a child's narrative macrostructure performance.*

**Hypothesis 4 (H4).** *Gender significantly predicts a child's narrative macrostructure performance.*

A hypothesized model that integrates the hypotheses was constructed, as shown in Figure 1.

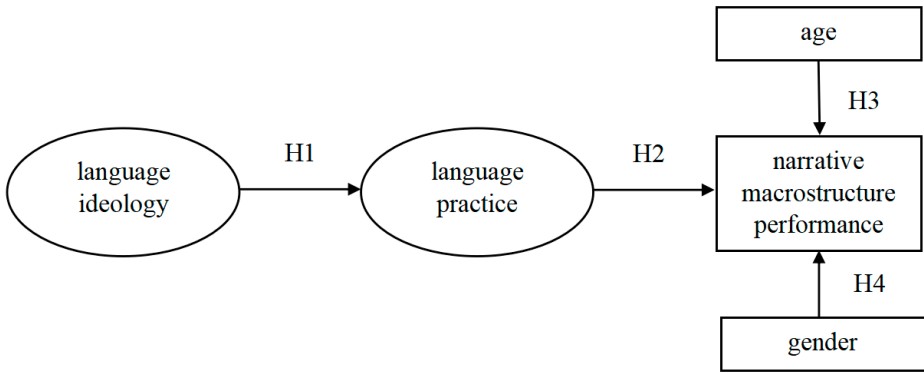

**Figure 1.** FLP and narrative macrostructure performance: the hypothesized model.

## 4. Methodology

### 4.1. Research Design

In order to explore the relationship between language ideology, language practice, and children's narrative macrostructure performance as well as the correlations with age and gender, the study utilized cross-sectional survey data collected through questionnaires and language performance test.

### 4.2. Participants

Participants were 131 Chinese-speaking children (64 males and 67 females, 4 to 7 years old) and their parents. The children were all born after the normal period of gestation and were healthy with normal intelligence. They were recruited from ten kindergartens located in the urban area of Linyi City, Shandong Province, China. Linyi is an average Chinese city in terms of economic and social development. The kindergartens, which belong to the same education group, all adopt the same standardized teaching management recommended by the Chinese government. Thus, the sample was quite representative of the pre-school children in China.

### 4.3. Measurement of Narrative Macrostructure Performance
4.3.1. Materials

Narrative macrostructure was evaluated by different ways, of which the most frequently used is the picture sequences complemented by standardized instructions [63,64], because compared to talking about one's own past experiences, the picture-prompted narrative tasks can elicit the shortest and the most cognitively demanding narratives [65]. Though some argued that telling a story with the help of a wordless picture book cannot reveal the narrative skills of children because the task happens in a conditioned way instead of daily conversation [66], many quantitative studies still adopted the picture sequence measurement, as it enables researchers to standardize the assessment of children's narrative skills and thus makes it possible to compare the performances of different children [64,67].

Therefore, we adopted the picture sequences measurement in the present study. The children were given a language performance test, in which children were asked to tell a story based on a wordless picture book titled "Frog, where are you?" by Mercer Mayer. Since both the length and the difficulty of the story are suitable for preschoolers, it is widely used by researchers worldwide, and its applicability for Chinese children was also confirmed [68].

### 4.3.2. Procedures

The test was carried out in the second week of the fall semester in 2019 to assess the children's development of narrative macrostructure. Each child was tested individually in a session that lasted around ten minutes. During the test, the children sat shoulder to shoulder with the researcher at a child-sided table in a quiet activity room which the children were very familiar with.

The researchers, who were careful not to describe the contents of the story, firstly introduced the children to the book and made sure that they understood the task. The test began when the children demonstrated that they knew what they were required to do. During the test, the researchers only helped the children to turn over the pages of the book and did not provide any suggestive or elicited utterances. When the test was over, the researchers expressed thanks and signaled the accomplishment of the task to the children by giving each of them a sticker as a gift.

### 4.3.3. Scoring

For the convenience of scoring, the whole procedure was recorded by recording pens. The recording material was first transcribed into text using a transcription software developed by Iflytek, a Chinese company specialized in intelligent speech and language technologies. The transcripts were then verified by two researchers manually to ensure the accuracy of the transcription.

The text was scored by one researcher and then checked by two other researchers from three aspects, i.e., plot lines (6 points), theme references (9 points), and misadventures (6 points) for both characters. Based on the description of the scoring criteria, which was set up by Miles and Chapman [69] and demonstrated in the picture book, a child would get 1 point if he mentioned 1 content described in the three aspects, and the total score of the narrative ability test was 21 points. A sum total of the scores in each aspect was regarded as the narrative macrostructure performance of a child.

### *4.4. Measurement of Language Ideology and Language Practice*

### 4.4.1. Materials

Language ideology and practice of the children's family were measured using a questionnaire survey. The questionnaire was designed and completed in Mandarin Chinese. All the items were translated into English by the first author and checked twice by the other two authors during paper writing. The questionnaire is composed of two parts. The first part is concerned with the demographical information, such as gender and age of the children and education level and occupation of their parents. The second part consists of Likert-scale questions measuring the families' language ideologies and language practices. Regarding language ideology, the respondents were asked to indicate whether they agreed with the following five statements on a 5-point Likert scale, with 1 meaning "strongly disagree" and 5 meaning "strongly agree":

LIa: Children's language ability is very important to their success in the future.
LIb: Oral expression is the most important language ability.
LIc: It is very important to have many children's books at home.
LId: It is very important to let children see their parents reading.
LIe: The language of family members exerts a huge influence on children.

With respect to language practice, the respondents were asked to indicate whether the following five descriptions were true of them on a 5-point Likert scale, with 1 meaning "not true of me at all" and 5 meaning "very true of me":

LPa: I read more than three times each week.
LPb: I often talk to my child about what happened in a day.
LPc: I often debate with my child.
LPd: I often repeat and extend what my child says.
LPe: I often induce my child to tell me stories.

### 4.4.2. Procedures

After the language performance test, the questionnaire survey was administered. The questionnaires were distributed to the parents when they came to pick the children up from kindergarten and were collected the next day when they sent their children back. One parent of each family, who was the major caregiver of their child, filled in the questionnaire, and they were told to inquire the researchers via Wechat or mobile phone whenever they met difficulties.

### 4.4.3. Data Entry

All the respondents filled in the questionnaires and handed them back to the kindergarten. Finally, 131 questionnaires were collected. One researcher entered the scores into SPSS 22.0 and the other two verified them to ensure the accuracy. During this process, if there was anything unclear or uncertain, the researchers discussed and marked it first and later consulted with the respondents through Wechat or mobile phone to make it clear, thus all the questionnaires were valid.

## 5. Results

### 5.1. Descriptive Statistics

Table 1 presents results of descriptive statistics of families' language ideology, language practice, as well as children's narrative macrostructure performance. It can be seen that the parents scored particularly high on the language ideology items (LIa, LIb, LIc, LId, LIe) and moderately high on the language practice items (LPa, LPb, LPc, LPd, LLe). The children's overall performances, with an average score of 10.71, which is lower than the 60% passing mark (12.60), were not satisfying.

**Table 1.** Language ideology, language practice, and narrative macrostructure performance.

|  | Mean | Std. Deviation | Range |
|---|---|---|---|
| LIa | 4.73 | 0.67 | 1–5 |
| LIb | 4.52 | 0.86 | 1–5 |
| LIc | 4.56 | 0.88 | 1–5 |
| LId | 4.65 | 0.80 | 1–5 |
| LIe | 4.67 | 0.70 | 1–5 |
| LPa | 3.56 | 1.26 | 1–5 |
| LPb | 4.08 | 1.10 | 1–5 |
| LPc | 3.69 | 1.20 | 1–5 |
| LPd | 3.85 | 1.03 | 1–5 |
| LPe | 4.25 | 0.88 | 1–5 |
| Narrative macrostructure score | 10.71 | 2.39 | 0–19 |

(Note: N = 131).

### 5.2. Model Evaluation

The hypothesized model was tested using structural equation modeling (SEM). As gender is an exogenous variable that is not predicted by any other variables (including latent variables) in the model, it does not require estimation methods specifically designed for categorical variables [70]. We thus assigned the value of 1 to males and the value of 2 to females in model estimation and used maximum likelihood as the estimation method. However, as the data deviated from multivariate normal distribution (multivariate kurtosis = 79.466; c.r. = 23.028), Bollen-Stine bootstrap correction, which is especially suitable for dealing with non-normal data with relatively small sample size, was performed to obtain more accurate results for model estimation [71]. The software used was AMOS 22.0.

The chi-square for the model was 79.693, with 64 degrees of freedom, a *p*-value of 0.089, and a Bollen–Stine correction p-value of 0.363. The *p*-values, which were larger than 0.05, indicated that the null hypothesis—that there was no significant difference between the covariance matrix of the sample and that hypothesized by the model—could not be

rejected. In other words, the model was plausible. Moreover, Table 2 shows a summary of model fit, from which it can be seen that the major model fit indices all fell into the suggested range, even by the strictest standards, suggesting a good fit [72,73].

**Table 2.** Model fit indices.

|  | CMIN/DF | CFI | IFI | TLI | RMSEA |
|---|---|---|---|---|---|
| Cut-off criteria | 1–2 | >0.95 | >0.95 | >0.95 | <0.06 |
| Actual values | 1.245 | 0.967 | 0.968 | 0.960 | 0.043 |

Figure 2 shows the standardized regression weights of each path in the model after Bollen-Stine correction, where * indicates that the path was significant ($p < 0.01$). In the measurement model, the ten factor loading values were all above 0.5, suggesting good construct validity.

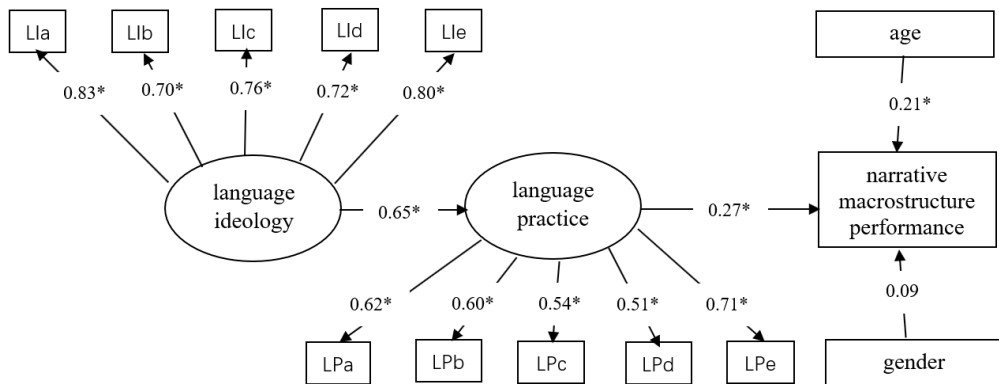

**Figure 2.** Relation between FLP and narrative macrostructure performance.

In the structural model, the path from language ideology to language practice, the path from language practice to narrative macrostructure performance, and the path from age to narrative macrostructure performance were significant. However, the path from gender to narrative macrostructure performance was not significant. In other words, of the four hypotheses raised, H1 (a family's language ideology significantly predicts its language practice), H2 (a family's language practice significantly predicts the child's narrative macrostructure performance), and H3 (age significantly predicts a child's narrative macrostructure performance) were verified, whereas H4 (gender significantly predicts a child's narrative macrostructure performance) was rejected.

To sum up, the results showed that a family's language ideology is a very strong predictor of its language practice, indicating that a family's language ideology exerts a huge influence on its language practice. In addition, a family's language practice and a child's age are significant predictors of the child's narrative macrostructure performance, whereas the child's gender does not affect his or her narrative macrostructure performance.

## 6. Discussion

All children, including those born into monolingual contexts, may have rich and varied language learning experiences. Early in this learning journey, children look to their parents or caregivers as their first teachers and role models, whose language practices are shaped by their own language beliefs. The present study not only corroborated the influence of family language ideology on language practice but located the relationship between language practice and the development of children' narrative ability as well, which provides us with references for future intervention.

### 6.1. Family Language Ideology Significantly Influences Family Language Practice

In alignment with language policy studies within the scope of dual language bilingual education (DLBE) programs in schools [19,20], the concept that family language ideology is the internal driving force of family language policy was visualized and confirmed by statistical analysis, which was consistent with the previous studies [4]. It can be clearly seen from the structural model that the path from language ideology to language practice was significant, which means a family's language practice is, to a large extent, shaped by its language ideology. Since kindergarteners are not cognitively mature enough to influence their parents, the family language ideology mainly comes from the parents' or the major caregivers' language ideologies or beliefs, which can be shaped by many linguistic and non-linguistic contexts, such as socio-economic context (national education or national language policy) [15]. This means if parents understand the value of language or language education, or if they are congruent with the national language policy or the national language education policy, they frame their understandings and map those understandings onto the family members and their language practices, and hence they have expectations for their children's language development and educational outcomes. Consequently, a successful FLP can be predicted.

Since family language ideology plays a crucial role in creating family literacy environments which facilitate children's language acquisition, parents should be informed of the results of research on narrative ability development through upper-level policies, various instructional approaches, or other forms so that they may have "better" language ideologies. For example, it can be expected parents will pay more attention to children's narrative ability development if they learn about the significant effect of kindergarten children's narrative skills on their emergent literacy development, successful adaptation to school literacy [74], and later school success [75,76].

### 6.2. Family Language Practice in Turn Shapes Children's Narrative Macrostructure Ability

The structural equation model demonstrated that family language practice could predict children's narrative macrostructure development, which pointed to the central role of family language practice in language acquisition. In particular, it showed that parents' own frequent reading activities and their actions of debating with their children, repeating and extending what their children said, and eliciting their children to tell stories all have positive effects on children's narrative macrostructure development. The forming of narrative ability is indispensable of language practices, thus it is parents and major caregivers who influence children's early learning, including language acquisition.

According to Spolsky, the observable language behaviors and language choices that occur in a social (family) interaction provide a natural context for language use and language learning, thus parents should be instructed to carry out language practices such as those investigated in the present study [77].

In addition, just as Carmiol and Sparks [78] pointed out, it is the internal driving force that children need to construct their narrative informatively for their listeners that helps form this ability, thus people first need to be attentive listeners and learners to stimulate this internal driving force in children in order to foster their narrative abilities. Successful ways of creating satisfying literacy environments for children should be investigated and summarized, and instructions should be made and open to society.

### 6.3. Development of Narrative Macrostructure Is a Predictable and Gradually Learned Process with Cognitive and Linguistic Demands

The results of the study indicate that, with regard to the narrative macrostructure development, differences between the age groups were significant, which was in line with the findings of previous studies on the narrative macrostructure development in children between four to six years old [30,44]. This also revealed that the forming of narrative abilities was a developmental procedure, which was based on cognitive and linguistic demands and could be fostered. Children gradually acquired this ability through observations and

experiences, and, during this course, the macrostructure of a story played a great role because it could provide an overall structure for enhancing children's understanding of story structure, especially when the story elements were relatively complicated. However, when narrating a story, young children might face difficulties. We could see this from our above analysis that children' mean score (10.71) in the performance test was lower than the pass marks (12.60). Therefore, parents and educators should take instructional measures to activate children cognitively and linguistically and try to formulate an interactive instructional context for children [79].

### 6.4. Development of Narrative Macrostructure Shows Weak Gender Effects

The results also indicated that the path from gender to narrative macrostructure performance was not significant, which meant H4 (gender significantly predicts a child's narrative macrostructure performance) was rejected. There are two possible explanations for the weak gender effects. One could be that the variables under study had little bearing on this hypothesis, and the other could be that the gender differences in narrative macrostructure development do not appear until late childhood. Previous studies mainly focused on genderlect (the language variety used by different sexes) in the narrative production, aiming at finding out gender-related differences. Little was done to testify whether there is a difference in narrative development between boys and girls. Therefore, for more knowledge of it, further study should be conducted.

### 7. Conclusions

This study provides a novel understanding of the crucial factors for sustainable development of early language education by testifying the relationship among language ideologies, language practices, and kindergartners' narrative macrostructure development. The study contributes to child language development research with the FLP framework in that it corroborates the relationship between two dimensions of FLP (language belief and language practice) as well as the relationship between language practice and children's narrative macrostructure development in a monolingual context. It also adds to discussion on the role of age and gender factors on narrative macrostructure development in that it reveals that narrative abilities change with age but not with gender. Moreover, the results provide implications for early intervention for children's narrative macrostructure development. That is, knowledge about the importance of children's early language development, particularly of narrative skills development, should be emphasized in language policy or language education policy making, and professional suggestions should be given to parents via different channels so that instructive interacting contexts could be created to guarantee children's narrative ability development, their sustainable development of early language education, and better adaptation to school work. In the future, data of early intervention should be collected to confirm its effectiveness and provide more pedagogical implications.

**Author Contributions:** J.Y. conceived the idea and wrote the paper. Y.D. analyzed the data and was involved in manuscript writing; L.F. supervised the study, reviewed and edited the writing. All authors have read and agreed to the published version of the manuscript.

**Funding:** This research was funded by National Social Science Fund of China, grant number 17BYY081 and 19AYY010, and the First-Class Disciplines Project of Beijing Foreign Studies University, grant number 2020SYLDXM040.

**Institutional Review Board Statement:** Not applicable.

**Informed Consent Statement:** Informed consent was obtained from all subjects involved in the study.

**Data Availability Statement:** The data presented in this study are available on request from the corresponding author.

**Conflicts of Interest:** The authors declare no conflict of interest.

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
