# Peer review of "Language Ideologies, Practices, and Kindergarteners’ Narrative Macrostructure Development: Crucial Factors for Sustainable Development of Early Language Education"

_sustainability, doi:10.3390/su13136985_

Round 1

Reviewer 1 Report

Line 28--delete- during  add- in which

Line 30--delete -so       add  ,therefore,

Line 63---add--  In other words, 

Line 71---delete paid sufficient attention------add   has not been a priority

Line 80---delete or    add   are

Line 81---add s    refers

Line 82--- delete  with    add  in

Line  102--delete individual person's    add  individual's personal

Line 106--- period after inquiries.  delete- though

Line 107---delete  paid     add  given

Line 110---delete  So far      add  Most

Line 134---remove coma after refined 

Line 135---add coma after old,      delete  Besides   add   Narrative

Line 147----delete  Up to date     add  To date,

Line 150----delete  grown up     add  growing up

Line 152-and Line 153--add   experiences, this will influence their

Line 156--and 157---remove   which is in turn- add--47, shaped by FLP.

Line 158----add coma after sharing,

Line 161--delete still    add  and others reported

Line 169-and 170-delete  in turn   add   which influences

Line 183--delete  so    add  therefore,

Line 184---remove  What is more    add   Most of the studies

Line  186--- delete  let alone     add   and the relationship

Line 188----add coma  after review,  add    it reveals

Line 236---delete  among      add    of  which

Line 424 ---delete learnt    add learners

Line 452---add    Seldom it has been done

Author Response

Point 1:

Line 28--delete- during  add- in whichLine 30--delete -so       add  ,therefore,

Line 63---add--  In other words, 

Line 71---delete paid sufficient attention------add   has not been a priority

Line 80---delete or    add   are

Line 81---add s    refers

Line 82--- delete  with    add  in

Line  102--delete individual person's    add  individual's personal

Line 106--- period after inquiries.  delete- though

Line 107---delete  paid     add  given

Line 110---delete  So far      add  Most

Line 134---remove coma after refined 

Line 135---add coma after old,      delete  Besides   add   Narrative

Line 147----delete  Up to date     add  To date,

Line 150----delete  grown up     add  growing up

Line 152-and Line 153--add   experiences, this will influence their

Line 156--and 157---remove   which is in turn- add--47, shaped by FLP.

Line 158----add coma after sharing,

Line 161--delete still    add  and others reported

Line 169-and 170-delete  in turn   add   which influences

Line 183--delete  so    add  therefore,

Line 184---remove  What is more    add   Most of the studies

Line  186--- delete  let alone     add   and the relationship

Line 188----add coma  after review,  add    it reveals

Line 236---delete  among      add    of  which

Line 424 ---delete learnt    add learners

Line 452---add    Seldom it has been done

Response 1: Thank you very much for your detailed comments on the manuscript. We have made all the changes as you suggested, except the following:

Line 82--- delete  with    add  inWe didn’t change the sentence, which was a direct quotation. We checked the original material again to make sure that the sentence was cited correctly. 

Reviewer 2 Report

This article empirically sheds light on the relationship between family language policy-language ideologies and language practices, which may have a positive impact on the development of children’s narrative macrostructure. The results also reveal some predictors of narrative macrostructure development that will surely be useful for further research on this domain. Furthermore, an adequate research method is provided. However, additional data on implications for early language intervention are required in order to amplify perspectives from a pedagogical point of view. Finally, this article must be carefully proofread (reference's style included). 

Author Response

Point 1: This article empirically sheds light on the relationship between family language policy-language ideologies and language practices, which may have a positive impact on the development of children’s narrative macrostructure. The results also reveal some predictors of narrative macrostructure development that will surely be useful for further research on this domain. Furthermore, an adequate research method is provided. However, additional data on implications for early language intervention are required in order to amplify perspectives from a pedagogical point of view. Finally, this article must be carefully proofread (reference's style included). 

Response 1: Thank you very much for your constructive comments. We all agree that additional data on implications for early language intervention are needed in order to confirm the effectiveness of the intervention and amply perspectives from a pedagogical point of view. However, unfortunately, we are not able to collect the data in the present study. We acknowledged the limitation, emphasized the need, and pointed out further research will be done in this aspect in follow-up studies. We also proofread the article (including references) again and corrected all the errors we found.

Reviewer 3 Report

The paper is of current relevance. The references are duly arranged.

The authors try to clearly visualizie the overall structure of the article.

However, it is recommended to improve the following aspects.

  1. The introduction provides the general outline of the concepts under study and sets forth the general description of gioals and results (lines 63-74). Meanwhile, the second section again starts with the focus on the Backgroundand adds the Literature Review item. It seems logical to avoid repetition and provide the background in the introduction..
  2. It might be useful to specify the research tasks and hypotheses in the intriduction after the statement on the research goal (to make it clear to the reader what the research essence is.)
  3. Now the research design, hypotheses and methods are not clearly balanced. Nor the sections of results and discussion  are coordinated in terms of logical coherence in line with the research hypotheses.
  4. The above put down the overall quality of structure and clarity. I would strogly recommend to coordinate in terms of wording and sense issues the headings and the contents of the mentioned sections. In its current version the paper lacks academic isomorphism regarding the structure and style of the data provision and its interpretation.

Author Response

Point 1: The introduction provides the general outline of the concepts under study and sets forth the general description of goals and results (lines 63-74). Meanwhile, the second section again starts with the focus on the Background and adds the Literature Review item. It seems logical to avoid repetition and provide the background in the introduction.

Response 1: Your comments and suggestions are highly appreciated. We all agree that we should avoid repetition, and we believe that we didn’t provide any background information in the second section. The original heading was misleading. So we deleted –Background in the heading of section 2. and left “Literature Review” only.  

Point 2: It might be useful to specify the research tasks and hypotheses in the introduction after the statement on the research goal (to make it clear to the reader what the research essence is.)

Response 2: Thank you very much for your suggestion. We specified the four research questions we investigated in the last part of the introduction. 

Point 3: Now the research design, hypotheses and methods are not clearly balanced. Nor the sections of results and discussions are coordinated in terms of logical coherence in line with the research hypotheses.

Response 3: Thank you for pointing this out. We reorganized the manuscript by putting Research hypothesis in Section 3, and methodology in Section 4. In section 4, we reorganized the contents into 4.1 Research design, 4.2 Participants, 4.3 Measurement of narrative macrostructure performance, and 4.4 Measurement of language ideology and language practice to make the research design, hypothesis and methods clearly balanced. In order to coordinate the sections of results and discussions in terms of logical coherence in line with the research hypothesis, in the results section we moved the narrative macrostructure score in Table 1 to the end of the list, and added the specific items of language ideology (LIa, LIb, LIc, Lid, LIe) and items of language practice (LPa, LPb, LPc, LPd, LPe) by putting them in parenthesis so that readers might understand Table 1 more easily. We also reorganize the discussion section by dividing it into four subsections, each discussing one hypothesis.  

Point 4: The above put down the overall quality of structure and clarity. I would strogly recommend to coordinate in terms of wording and sense issues the headings and the contents of the mentioned sections. In its current version the paper lacks academic isomorphism regarding the structure and style of the data provision and its interpretation.

Response 4: Thank you for your kind comments and suggestions. We have made the above-mentioned revisions regarding the structure and style of data provision and its interpretation to improve academic isomorphism.